# Utility of hospitalization for elderly individuals affected by COVID-19

**Giorgio Costantino[1,2], Monica Solbiati** [1,2]*, **Silvia Elli[3], Marco Paganuzzi[3], Didi Massabò[3], Nicola Montano[1,2], Marta Mancarella[1], Francesca Cortellaro[4], Emanuela Cataudella[4], Andrea Bellone[5], Nicolò Capsoni[5], Guido Bertolini** [6], **Giovanni Nattino[6], Giovanni Casazza[7]**

**1** Fondazione IRCCS Ca' Granda Ospedale Maggiore Policlinico, Milan, Italy, **2** Dipartimento di Scienze Cliniche e di Comunità, Università degli Studi di Milano, Milan, Italy, **3** Università degli Studi di Milano, Milan, Italy, **4** ASST Santi Paolo e Carlo, Pronto Soccorso e Degenza Breve San Carlo, Milan, Italy, **5** ASST Grande Ospedale Metropolitano Niguarda, Medicina d'Urgenza e Pronto Soccorso, Milan, Italy, **6** Laboratorio di Epidemiologia Clinica, Dipartimento di Salute Pubblica, Istituto di Ricerche Farmacologiche Mario Negri IRCCS, Bergamo, Italy, **7** Dipartimento di Scienze Biomediche e Cliniche "L. Sacco", Università degli Studi di Milano, Milan, Italy

* monica.solbiati@gmail.com

## Abstract

### Background

During the COVID-19 pandemic, the number of individuals needing hospital admission has sometimes exceeded the availability of hospital beds. Since hospitalization can have detrimental effects on older individuals, preference has been given to younger patients. The aim of this study was to assess the utility of hospitalization for elderly affected by COVID-19. We hypothesized that their mortality decreases when there is greater access to hospitals.

### Methods

This study examined 1902 COVID-19 patients consecutively admitted to three large hospitals in Milan, Italy. Overall mortality data for Milan from the same period was retrieved. Based on emergency department (ED) data, both peak and off-peak phases were identified. The percentage of elderly patients admitted to EDs during these two phases were compared by calculating the standardized mortality ratio (SMR) of the individuals younger than, versus older than, 80 years.

### Results

The median age of the patients hospitalized during the peak phase was lower than the median age during the off-peak phase (64 vs. 75 years, respectively; p <0.001). However, while the SMR for the younger patients was lower during the off-peak phase (1.98, 95% CI: 1.72–2.29 versus 1.40, 95% CI: 1.25–1.58, respectively), the SMR was similar between both phases for the elderly patients (2.28, 95% CI: 2.07–2.52 versus 2.48, 95% CI: 2.32–2.65, respectively).

**Data Availability Statement:** All relevant data are within the manuscript and its Supporting Information files.

**Funding:** This research was funded from the Tsunami project of the Lombardy region. The funders had no role in study design, data collection and analysis, decision to publish, or preparation of the manuscript.

**Competing interests:** The authors have declared that no competing interests exist.

## Conclusions

Greater access to hospitals during an off-peak phase did not affect the mortality rate of COVID-19-positive elderly patients in Milan. This finding, if confirmed in other settings, should influence future decisions regarding resource management of health care organizations.

## Introduction

The COVID-19 pandemic has had tremendous consequences on the Lombardy health care system [1,2]. A key challenge has been the simultaneous admission of patients who manifest respiratory failure to emergency departments (EDs). These patients require supplemental oxygen, ventilation, or endotracheal intubation [3]. As a result, there has been a greater need for hospital and intensive care unit beds for patients admitted with COVID-19 than there are resources available [4]. In addition, many patients have not been able to gain access to hospitals because the number of requests for help exceed the availability of ambulances to provide emergency medical services [5]. In particular, the population of elderly patients has most often been affected by this scarcity of resources. Indeed, during the peak of the current COVID-19 epidemic, older individuals represented a smaller proportion of in-hospital patients than during off-peak periods. It would be predicted that obstacles to access of health care facilities would increase the rates of mortality for the elderly more than for younger individuals during the peak of the epidemic. However, there is no evidence that hospitalization positively affects the prognosis of older individuals, particularly those with conditions which provoke severe acute respiratory failure (e.g., COVID-19-associated pneumonia) [6,7]. Indeed, older individuals are at higher risk of serious outcomes in case of infections and it is not known whether hospital admission and aggressive treatments are useful in such patients [8].As some countries are still dealing with the epidemic peak, and others are preparing to face a possible second peak, it is important to understand whether hospital admission is beneficial for the elderly population. This is especially relevant when critical resources need to be efficiently allocated. Therefore, the aim of this study was to assess the utility of hospitalization for elderly individuals affected by COVID-19. We hypothesized that the mortality rates for elderly individuals are higher during peak phases of the epidemic than during off-peak phases when greater access to hospitals is available.

## Materials and methods

### Study design

An observational multicenter study was conducted with data obtained from ED databases and population mortality reports.

### Study setting and population

Two sources of information were collected: 1) data for COVID-19-positive patients in Milan who were admitted to major hospitals from an ED and 2) population mortality data for Milan from March 1 to April 30, 2020. The latter interval was selected because it represents a peak epidemic period in Milan. All of the patients who presented to EDs at the following three major hospitals in Milan were included: Fondazione IRCCS Ca' Granda Ospedale Maggiore Policlinico, ASST Grande Ospedale Metropolitano Niguarda, and Ospedale San Carlo Borromeo. These hospitals account for approximately 250,000–300,000 admittances to EDs each year. The above databases were accessed on June 16th, 2020.

## Measures

The daily numbers of patients who presented to EDs with COVID-19 were analyzed to identify peak and non-peak phases of the COVID-19 epidemic. A peak phase was defined as a seven-day period with the highest mean number of COVID-19 patients presenting to EDs. An off-peak phase was defined as the 14-day period starting 14 days after the end of the 7-day peak. The length of the latter period was selected in order to have a similar number of patients enrolled for both phases.

From the ED databases of the three hospitals, the following data were extracted for all of the patients included in this study: date of presentation at ED, age, and gender. Age and gender were compared between the peak and off-peak phases. In addition, the patients examined were divided into two age groups, < 80 years and ≥ 80 years.

Mortality data released by the Italian National Institute of Statistics (ISTAT) and for the general population of Milan between January 1, 2015 and April 30, 2020 were analyzed [9]. In particular, the overall numbers of daily deaths according to age, with 5-year age classes established, were extracted. Initially, a descriptive analysis of the number of deaths per day for the two age groups was conducted. Next, cumulative mortality for the peak and off-peak phases for the two age groups (< 80 y and ≥ 80 y) was calculated. In order to compare observed mortality between the age groups and phases, a standardized mortality ratio (SMR) was calculated by dividing the number of observed deaths by the mean number of deaths observed from 2015 to 2019. SMRs for the peak and off-peak phases were also calculated separately for the two age groups. Finally, mortality during the peak and off-peak phases was compared by calculating the ratio between the corresponding SMRs. Assuming a Poisson distribution for the deaths in 2020 and that the expected number of deaths (mean number of deaths from 2015 to 2019) were error free, 95% confidence intervals (CI) were calculated for the SMRs. We also calculated 95% CIs for the ratio of the two SMRs according to the procedure described by Ederer [10].

## Data analysis

All statistical analyses were performed with the statistical software, SAS (release 9.4, SAS Institute, Inc., Cary, NC, USA). All patients' data were anonymized before the researchers accessed the data and the Ethics Committee of the coordinating center (Fondazione IRCCS Ca' Granda Ospedale Maggiore Policlinico) approved the protocol for this study. Continuous variables are reported as median values (interquartile range, IQR), while categorical data are reported as counts (percentages). Characteristics of the patients who presented to an ED during the two time-periods were compared by applying Fisher's exact test to the categorical variables, and the Wilcoxon two-sample test to the continuous variables. Sensitivity analyses were performed considering different age subgroups. Two-sided p-values less than 0.05 were considered statistically significant.

## Results

Between March 1 and April 30, 2020, a total of 1902 COVID-19-positive patients were admitted to three major hospitals in Milan. A peak phase of the COVID-19 epidemic was identified between March 16 and March 22, 2020 when a total of 396 patients were admitted. This corresponds to a mean admission of 56.6 patients per day. An off-peak phase was subsequently identified between April 6 and April 19, 2020. During the latter phase, a total of 338 patients were admitted, with a mean admission of 24.1 patients per day. Table 1 reports the characteristics of the patients examined. The patients admitted during the peak phase had a lower median age than the patients admitted during the off-peak phase.

Table 2 presents overall mortality data for the two time-periods examined.

**Table 1. Characteristics of the COVID-19-positive patient population examined.**

|  |  | Peak (n = 396) | Off-peak (n = 338) |  |
|---|---|---|---|---|
| Age | < 80 y | 335 (84.8%) | 206 (61.0%) | P < 0.0001[a] |
|  | ≥ 80 y | 60 (15.2%) | 132 (39%) |  |
| Median age (IQR) |  | 64 (54 to 75) | 74.5 (56 to 83) | P ≤ 0.0001[b] |
| Gender | Female | 124 (31.3%) | 166 (49.1%) | P ≤ 0.0001[a] |
|  | Male | 272 (68.7%) | 172 (50.9%) |  |

IQR: Interquartile range

[a] Fisher's exact test

[b] Wilcoxon two-sample test.

The data presented in Fig 1 show that the mortality trend over time is similar for the two age groups.

We observed that standardized mortality among the younger patients was higher during the peak phase than during the off-peak phase (SMR: 1.98, 95% CI: 1.72 to 2.29 vs. SMR: 1.40, 95% CI: 1.25 to 1.58, respectively). In contrast, standardized mortality for the older patients was similar for the two phases examined (SMR: 2.28, 95% CI: 2.07 to 2.52 vs. SMR: 2.48, 95% CI: 2.32 to 2.65, respectively). Sensitivity analyses considering different age subgroups confirmed these results (Table 3).

## Discussion

The aim of this study was to examine whether decreased access to hospitals due to limited emergency medical services transportation in Milan affected the mortality of older individuals during the recent COVID-19 epidemic. Indeed, public opinion criticized the decision not to take elderly individuals to hospitals (especially those coming from nursing homes). The corresponding ethical consequences were also discussed. During the peak phase, we observed a lower number of very elderly patients, and a lower median age of admitted people, when compared with the off-peak phase. This is the demonstration that ED access of very elderly people was reduced during the peak phase. Our data confirm that the rate of hospitalization for patients older than 80 years was lower during the peak phase of the epidemic than during an off-peak phase. However, mortality did not differ between the two periods for these elderly patients. In contrast, the off-peak mortality for younger individuals was significantly lower than the mortality during the peak phase because of a decrease in the number infected patients. Taken together, these findings suggest that the mortality of elderly COVID-19-positive patients is related to the disease itself, and not to a lack of proper care (i.e., hospitalization). Thus, limited access to hospitalization did not increase mortality in patients older than 80 years during the recent COVID-19 epidemic in Milan.

**Table 2. Number of expected and observed deaths during both peak and off-peak phases of the COVID-19 epidemic in Milan.**

|  | Number of expected deaths | | Number of observed deaths | |
|---|---|---|---|---|
|  | Peak (n) | Off-peak (n) | Peak (n) | Off-peak (n) |
| **All age groups** | **265.8** | **531.6** | **578** | **1119** |
| ≥ 85 y | 124.8 | 257 | 270 | 652 |
| 80–84 y | 46.2 | 89 | 120 | 207 |
| 70–79 y | 58.4 | 102.6 | 125 | 163 |
| < 70 y | 36.4 | 83 | 63 | 97 |

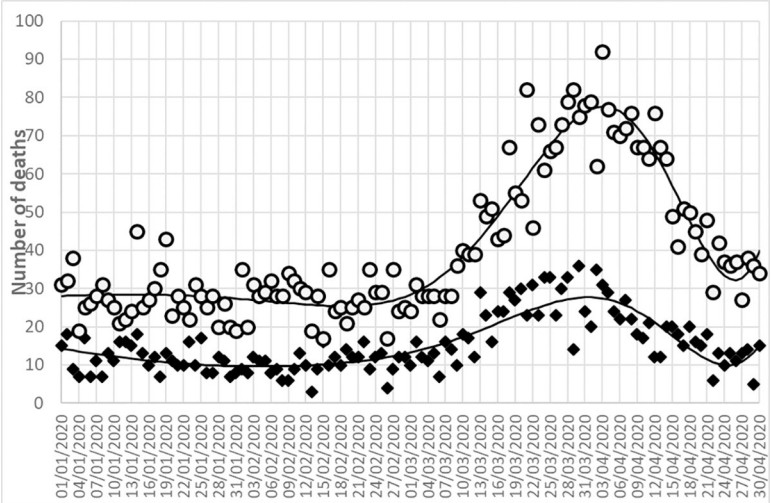

**Fig 1. Number of deaths in 2020 according to age class.** White dots represent the daily deaths of patients aged $\geq 80$ y, black diamonds represent the daily deaths of patients $< 80$ y. Solid lines represent trends among the data.

This finding, which disproves our initial hypothesis, may be explained in different ways. First, at the time the study refers to, no therapy had proved to be effective for COVID-19, except for supportive respiratory measures [11–17] and the currently available evidence on the possible efficacy of steroids, Remdesivir and heparin was lacking [12,18,19]. The greater predictor of survival has been found to be the medical history and health status of patients, as well as age and a lack of comorbidities [20,21]. For example, while supportive respiratory measures during hospitalization may be useful for providing oxygen and ventilation to younger patients until they heal naturally, these supportive measures may not be sufficient for the elderly. Second, hospitalization of older patients may have involved less intensive measures than those administered to younger patients. For example, elderly patients are usually not candidates for ICU admission, and they tend to be treated less frequently with non-invasive ventilation. Finally, hospitalization itself can have dramatic consequences and increase morbidity and mortality as a result of higher rates of complications such as nosocomial infections, delirium, bed confinement, and thrombosis [22].

Our results have several implications for future decisions regarding health care organization during the COVID-19 epidemic. One implication is that in a setting of limited resources and

**Table 3. Standardized mortality according to age during peak and off-peak phases of the COVID-19 epidemic in Milan.**

|  | Peak SMR (95% CI) | Off-peak SMR (95% CI) | Peak/Off-peak Ratio of SMRs (95% CI) |
|---|---|---|---|
| All age groups | 2.17 (2.00 to 2.36) | 2.10 (1.99 to 2.23) | 1.03 (0.93 to 1.14) |
| $\geq 80$ y | 2.28 (2.07 to 2.52) | 2.48 (2.32 to 2.65) | 0.92 (0.81 to 1.04) |
| $< 80$ y | 1.98 (1.72 to 2.29) | 1.40 (1.25 to 1.58) | 1.41 (1.17 to 1.72) |
| **Sensitivity analysis** | **Peak SMR (95% CI)** | **Off-peak SMR (95% CI)** | **Peak/Off-peak Ratio of SMRs (95% CI)** |
| $\geq 85$ y | 2.16 (1.92 to 2.44) | 2.54 (2.35 to 2.74) | 0.85 (0.74 to 0.98) |
| $< 70$ y | 1.73 (1.35 to 2.22) | 1.17 (0.96 to 1.43) | 1.48 (1.06 to 2.05) |
| 65–79 y | 2.10 (1.78 to 2.47) | 1.57 (1.37 to 1.80) | 1.34 (1.07 to 1.67) |

SMR: Standardized mortality ratio; CI: Confidence interval.

Sensitivity analyses with patients in different age categories were also conducted.

increased demand, hospitalization of elderly patients does not appear to affect their prognosis. Rather, hospitalization was found to worsen their quality of life and increase the overcrowding of health care facilities. Therefore, our data indicate that the focus for COVID-19-positive elderly patients should be providing healthcare, and possibly palliative care, outside of a hospital setting (i.e., at home).

While the data examined apply to the current COVID-19 epidemic, it is possible that the observation that hospitalization does not positively affect elderly patients' prognosis may be true for other diseases, especially where effective therapy is lacking. The Choosing Wisely campaign has recently highlighted the need to reconsider the benefit(s) of hospital admission [23]. Indeed, over the past few years, the medical community has been more supportive of the concept that doing more does not always mean doing better. Furthermore, some interventions and investigations may cause more harm than good [24]. There are several studies which have demonstrated that hospitalization is unlikely to affect patients' prognosis in many conditions, such as syncope and pneumonia. Correspondingly, certain scientific societies have advised healthcare professionals to consider the risk of hospital admission when developing a patient's management strategy [25].

## Limitations

There are limitations associated with the current investigation. First, we considered "instant" deaths in the peak phases without considering that some deaths could be delayed. However, since we analyzed mortality as the ratio between the two age groups in the two periods, the effects of this potential bias should have been mitigated. Moreover, the mortality in May (two weeks after the off-peak phase) was in line with previous years [9]. Moreover, in the absence of data regarding the incidence of COVID-19 infections, the observed similarity in the rates of elderly mortality between the peak and off-peak periods of the recent COVID-19 epidemic in Milan may be due to differences in the timing of infection in the elderly. However, we observed that the peak in mortality for younger patients only differed by two days from the peak in older patients. Thus, we consider it unlikely that these two populations were infected at different times. A second limitation of the present study is that it is an observational study. Ideally, a randomized controlled trial should be conducted to validate the present findings. However, it is difficult to perform this type of trial within the current context of this epidemic. Moreover, we considered overall mortality irrespective of the setting (in-hospital vs out-of-hospital) and the cause of death. However, comparing mortality of hospitalized and non-hospitalized patients would be misleading, as patients admitted to hospital have usually a more severe clinical presentation. In addition, mortality data in Italy do not distinguish between in hospital and out of hospital deaths, so overall mortality includes also patients who died in hospital. Analyzing only patients who died of COVID-19 would ideally be better, but we preferred to assess overall mortality because death certificates are sometimes inaccurate in reporting the death cause, and also many patients could have had COVID-19 without being tested, and the true mortality due to SARS-CoV-2 infection might be underestimated.

Finally, since the administrative nature of the data, we had no availability of other information useful for stratified analyses, such as determinants of mortality beyond age (e.g. frailty) and place of death (e.g. in-hospital, out-of-hospital).

## Conclusions

The results of this study show that hospitalization did not affect the mortality of older COVID-19-positive patients during the recent COVID-19 epidemic in Milan. This finding, if

confirmed in other settings, should be considered by health care organizations and may influence their decisions regarding optimization of available resources. For example, intervention for treating older patients at home or in nursing homes might help a better allocation of resources while providing them the best care.

## Supporting information

**S1 Table. Anonymized data set.**
(XLSX)

## Acknowledgments

The authors would like to acknowledge the Fenice Network for their help.

## Author Contributions

**Conceptualization:** Giorgio Costantino, Monica Solbiati, Didi Massabò, Nicola Montano, Andrea Bellone, Guido Bertolini, Giovanni Nattino, Giovanni Casazza.

**Data curation:** Giorgio Costantino, Silvia Elli, Marco Paganuzzi, Didi Massabò, Marta Mancarella, Francesca Cortellaro, Emanuela Cataudella, Nicolò Capsoni, Giovanni Nattino, Giovanni Casazza.

**Formal analysis:** Monica Solbiati, Silvia Elli, Guido Bertolini, Giovanni Nattino, Giovanni Casazza.

**Funding acquisition:** Giorgio Costantino, Guido Bertolini.

**Investigation:** Giorgio Costantino, Monica Solbiati, Silvia Elli, Marco Paganuzzi, Didi Massabò, Marta Mancarella, Francesca Cortellaro, Emanuela Cataudella, Nicolò Capsoni, Guido Bertolini, Giovanni Casazza.

**Methodology:** Giorgio Costantino, Monica Solbiati, Marco Paganuzzi, Andrea Bellone, Guido Bertolini, Giovanni Nattino, Giovanni Casazza.

**Project administration:** Giorgio Costantino, Didi Massabò, Giovanni Casazza.

**Resources:** Giorgio Costantino.

**Software:** Monica Solbiati, Giovanni Casazza.

**Supervision:** Giorgio Costantino, Monica Solbiati, Francesca Cortellaro, Andrea Bellone, Guido Bertolini, Giovanni Casazza.

**Validation:** Giorgio Costantino, Silvia Elli, Marco Paganuzzi, Didi Massabò, Guido Bertolini, Giovanni Nattino, Giovanni Casazza.

**Visualization:** Giorgio Costantino, Marco Paganuzzi, Nicola Montano, Marta Mancarella, Nicolò Capsoni, Guido Bertolini, Giovanni Nattino, Giovanni Casazza.

**Writing – original draft:** Giorgio Costantino, Monica Solbiati, Giovanni Casazza.

**Writing – review & editing:** Silvia Elli, Marco Paganuzzi, Didi Massabò, Nicola Montano, Marta Mancarella, Francesca Cortellaro, Emanuela Cataudella, Andrea Bellone, Nicolò Capsoni, Guido Bertolini, Giovanni Nattino.

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
