## [Decision Letter · Decision Letter 0]

18 Mar 2021

PONE-D-21-06946

Utility of hospitalization for elderly individuals affected by COVID-19

PLOS ONE

Dear Dr.ssa Solbiati,

Thank you for submitting your manuscript to PLOS ONE. After careful consideration, we feel that it has merit but does not fully meet PLOS ONE’s publication criteria as it currently stands. Therefore, we invite you to submit a revised version of the manuscript that addresses the points raised during the review process.

We look forward to receiving your revised manuscript.

Kind regards,

Francesco Di Gennaro

Academic Editor

PLOS ONE

Journal Requirements:

Thank you for providing the date(s) when patient medical information was initially recorded. Please also include the date(s) on which your research team accessed the databases/records to obtain the retrospective data used in your study.

Thank you for stating in the text of your manuscript "All patients’ data were anonymized". Please clarify whether all patient data was anonymized before the researchers accessed the data. Please also add this information to your ethics statement in the online submission form.

We note that you have indicated that data from this study are available upon request. PLOS only allows data to be available upon request if there are legal or ethical restrictions on sharing data publicly. For information on unacceptable data access restrictions, please see http://journals.plos.org/plosone/s/data-availability#loc-unacceptable-data-access-restrictions.

4a) If there are ethical or legal restrictions on sharing a de-identified data set, please explain them in detail (e.g., data contain potentially identifying or sensitive patient information) and who has imposed them (e.g., an ethics committee). Please also provide contact information for a data access committee, ethics committee, or other institutional body to which data requests may be sent.

4b) If there are no restrictions, please upload the minimal anonymized data set necessary to replicate your study findings as either Supporting Information files or to a stable, public repository and provide us with the relevant URLs, DOIs, or accession numbers. Please see http://www.bmj.com/content/340/bmj.c181.long for guidelines on how to de-identify and prepare clinical data for publication. For a list of acceptable repositories, please see http://journals.plos.org/plosone/s/data-availability#loc-recommended-repositories.

Thank you for stating the following in the Acknowledgments Section of your manuscript:

'The authors would like to acknowledge the Fenice Network for their support.'

'This research was funded from the Tsunami project of the Lombardy region. The funders had no role in study design, data collection and analysis, decision to publish, or preparation of the manuscript.'

b. Please include the updated Funding Statement in your cover letter. We will change the online submission form on your behalf.

6. One of the noted authors is a group or consortium; the Fenice network.

In addition to naming the author group, please list the individual authors and affiliations within this group in the acknowledgments section of your manuscript.

Please also indicate clearly a lead author for this group along with a contact email address.

Additional Editor Comments:

dear Authors follow reviewer suggestions to improve your paper

Reviewers' comments:

Reviewer's Responses to Questions

**Comments to the Author**

1. Is the manuscript technically sound, and do the data support the conclusions?

Reviewer #1: Partly

Reviewer #2: Yes

2. Has the statistical analysis been performed appropriately and rigorously? 

Reviewer #1: I Don't Know

Reviewer #2: Yes

3. Have the authors made all data underlying the findings in their manuscript fully available?

Reviewer #1: Yes

Reviewer #2: Yes

4. Is the manuscript presented in an intelligible fashion and written in standard English?

Reviewer #1: Yes

Reviewer #2: Yes

5. Review Comments to the Author

Reviewer #1: the data does not address the difference in mortality for the elderly people admitted to the hospital versus those who were not admitted. If the SMR was lower for those hospital admitted compared to those not admitted to the hospitals then there is a benefit of hospitalization. I would consider the data valid if it compared in and out of hospital patients for both groups during the peak and off peak periods. the complexity of the disease and the lack of knowledge about its effect on different organs and thus treatment options makes it too simplified to compare peak versus off peak in the two groups. Also, it would be of value if there is comparison with similar data or studies from other counties or areas in the same country

Reviewer #2: Authors wrote an very interesting paper with focus on elderly. Knowledge on SARS CoV2 is ongoing and this paper can be contribute to increase the knowledge and awarness on this pandemic.

Below my suggestions:

1. Introduction: update data on burden of SARS CoV2 at the day of resubmission and add why for you elderly is a very important and crucial group. Elderly are most vulnerable population on infection (see and cite Active Pulmonary Tuberculosis in Elderly Patients: A 2016-2019 Retrospective Analysis from an Italian Referral Hospital. Antibiotics). In fact, The geriatric population in a high income setting such as in Italy represents a risk factor group for worste outcome of all infectious dieseaes (TB, COVID 19 etc) across all sexual and gender subgroups. Age-related comorbidities (e.g., malnutrition, cancer, chronic renal failure, and diabetes mellitus), together with physiological biological changes may weaken protective barriers, impair microbial clearance mechanisms, and contribute to reducing cellular immune responses

Methods and results: are very clear. No suggestion

Discussion: compare your data with other big data from other study on comorbidities (Common cardiovascular risk factors and in-hospital mortality in 3,894 patients with COVID-19: survival analysis and machine learning-based findings from the multicentre Italian CORIST Study. Nutr Metab Cardiovasc Dis. 2020 Oct 30;30(11):1899-1913), role of Heparin and treatment (Heparin in COVID-19 Patients Is Associated with Reduced In-Hospital Mortality: the Multicenter Italian CORIST Study. Thromb Haemost. 2021)

Conclusion: please add one two public health proposal that came from your very interesting paper

6. PLOS authors have the option to publish the peer review history of their article (what does this mean?). If published, this will include your full peer review and any attached files.

Reviewer #1: **Yes: **Aly Ahmed Abel Rahim

Reviewer #2: No

---

## [Author Response · Author response to Decision Letter 0]

28 Mar 2021

Journal Requirements:

Thank you for this suggestion. We checked that the manuscript meets PLOS ONE's style requirements.

2. Thank you for providing the date(s) when patient medical information was initially recorded. Please also include the date(s) on which your research team accessed the databases/records to obtain the retrospective data used in your study.

Thank you for this suggestion. We added the date on which we accessed the databases in the Materials and methods section.

3. Thank you for stating in the text of your manuscript "All patients’ data were anonymized". Please clarify whether all patient data was anonymized before the researchers accessed the data. Please also add this information to your ethics statement in the online submission form.

Thank you for this suggestion. We added the date on which we accessed the databases in the Materials and methods section.

4a) If there are ethical or legal restrictions on sharing a de-identified data set, please explain them in detail (e.g., data contain potentially identifying or sensitive patient information) and who has imposed them (e.g., an ethics committee). Please also provide contact information for a data access committee, ethics committee, or other institutional body to which data requests may be sent.

4b) If there are no restrictions, please upload the minimal anonymized data set necessary to replicate your study findings as either Supporting Information files or to a stable, public repository and provide us with the relevant URLs, DOIs, or accession numbers. Please see http://www.bmj.com/content/340/bmj.c181.long for guidelines on how to de-identify and prepare clinical data for publication. For a list of acceptable repositories, please see http://journals.plos.org/plosone/s/data-availability#loc-recommended-repositories.

Thank you for the opportunity to clarify. As there are no ethical or legal restrictions on sharing a de-identified data set, we have now uploaded the anonymized data set as a supporting information xls file. We have also updated the Data Availability statement on the submission system.

'The authors would like to acknowledge the Fenice Network for their support.'

Thank you for the opportunity to clarify. The above statement was probably misleading. We meant that we got help and not funding from the Fenice Network. We modified the statement in the acknowledgments accordingly.

'This research was funded from the Tsunami project of the Lombardy region. The funders had no role in study design, data collection and analysis, decision to publish, or preparation of the manuscript.'

b. Please include the updated Funding Statement in your cover letter. We will change the online submission form on your behalf.

Thank you for the opportunity to clarify. The funding statement is correct. We added it to the cover letter as suggested.

6. One of the noted authors is a group or consortium; the Fenice network.

In addition to naming the author group, please list the individual authors and affiliations within this group in the acknowledgments section of your manuscript.

Please also indicate clearly a lead author for this group along with a contact email address.

Thank you for the opportunity to clarify. As the Fenice network is a group of emergency departments and not of people, we agreed to remove the group from the authors and leave it only in the Acknowledgment section of the manuscript.

Additional Editor Comments:

dear Authors follow reviewer suggestions to improve your paper

Reviewers' comments:

Reviewer's Responses to Questions

Comments to the Author

5. Review Comments to the Author

Reviewer #1: the data does not address the difference in mortality for the elderly people admitted to the hospital versus those who were not admitted. If the SMR was lower for those hospital admitted compared to those not admitted to the hospitals then there is a benefit of hospitalization. I would consider the data valid if it compared in and out of hospital patients for both groups during the peak and off peak periods. the complexity of the disease and the lack of knowledge about its effect on different organs and thus treatment options makes it too simplified to compare peak versus off peak in the two groups. Also, it would be of value if there is comparison with similar data or studies from other counties or areas in the same country.

Thank you for this comment. We agree that our work has some limitations. However, comparing mortality of hospitalized and non-hospitalized patients would be misleading, as patients admitted to hospital have usually a more severe clinical presentation. As it would be unethical to randomize, we chose to compare peak and off peak periods because access to hospital was different between the two pandemic phases for organizational reasons. In addition, mortality data in Italy do not distinguish between in hospital and out of hospital deaths, so overall mortality includes also patients who died in hospital. Knowing the cause of death would probably be of help, but we preferred to use overall mortality because death certificates are sometimes inaccurate in reporting the death cause, and also many patients could have had COVID-19 without being tested, and the true mortality due to SARS-CoV-2 infection might be underestimated.

We tried to better acknowledge all the above considerations and added a sentence to the limitations section.

Reviewer #2: Authors wrote an very interesting paper with focus on elderly. Knowledge on SARS CoV2 is ongoing and this paper can be contribute to increase the knowledge and awarness on this pandemic.

Below my suggestions:

1. Introduction: update data on burden of SARS CoV2 at the day of resubmission and add why for you elderly is a very important and crucial group. Elderly are most vulnerable population on infection (see and cite Active Pulmonary Tuberculosis in Elderly Patients: A 2016-2019 Retrospective Analysis from an Italian Referral Hospital. Antibiotics). In fact, The geriatric population in a high income setting such as in Italy represents a risk factor group for worste outcome of all infectious dieseaes (TB, COVID 19 etc) across all sexual and gender subgroups. Age-related comorbidities (e.g., malnutrition, cancer, chronic renal failure, and diabetes mellitus), together with physiological biological changes may weaken protective barriers, impair microbial clearance mechanisms, and contribute to reducing cellular immune responses

Methods and results: are very clear. No suggestion

Discussion: compare your data with other big data from other study on comorbidities (Common cardiovascular risk factors and in-hospital mortality in 3,894 patients with COVID-19: survival analysis and machine learning-based findings from the multicentre Italian CORIST Study. Nutr Metab Cardiovasc Dis. 2020 Oct 30;30(11):1899-1913), role of Heparin and treatment (Heparin in COVID-19 Patients Is Associated with Reduced In-Hospital Mortality: the Multicenter Italian CORIST Study. Thromb Haemost. 2021)

Conclusion: please add one two public health proposal that came from your very interesting paper

Thank you for this comment. We modified the introduction, discussion and conclusion section as suggested.

---

## [Decision Letter · Decision Letter 1]

13 Apr 2021

Utility of hospitalization for elderly individuals affected by COVID-19

PONE-D-21-06946R1

Dear Dr.ssa Solbiati,

We’re pleased to inform you that your manuscript has been judged scientifically suitable for publication and will be formally accepted for publication once it meets all outstanding technical requirements.

Kind regards,

Francesco Di Gennaro

Academic Editor

PLOS ONE

Additional Editor Comments (optional):

congratulations

Reviewers' comments:

Reviewer's Responses to Questions

**Comments to the Author**

1. If the authors have adequately addressed your comments raised in a previous round of review and you feel that this manuscript is now acceptable for publication, you may indicate that here to bypass the “Comments to the Author” section, enter your conflict of interest statement in the “Confidential to Editor” section, and submit your "Accept" recommendation.

Reviewer #1: (No Response)

Reviewer #2: All comments have been addressed

2. Is the manuscript technically sound, and do the data support the conclusions?

Reviewer #1: Yes

Reviewer #2: Yes

3. Has the statistical analysis been performed appropriately and rigorously? 

Reviewer #1: Yes

Reviewer #2: Yes

4. Have the authors made all data underlying the findings in their manuscript fully available?

Reviewer #1: Yes

Reviewer #2: No

5. Is the manuscript presented in an intelligible fashion and written in standard English?

Reviewer #1: Yes

Reviewer #2: Yes

6. Review Comments to the Author

Reviewer #1: thank you for your response. I do understand the difficulty in making a valid comparison between in and out patients mortality. however if you consider making a comparison based on the available data. for example percentage of hospital death and percentage of outpatient death in the peak and non peak time and if there is difference death rate in both time period

Reviewer #2: Authors wrote an important paper. This new version, in my opinoin can be publish

Sharing the experience on SARS CoV2 is fundamental to control this pandemic

7. PLOS authors have the option to publish the peer review history of their article (what does this mean?). If published, this will include your full peer review and any attached files.

Reviewer #1: **Yes: **Aly Ahmed Abdel Rahim

Reviewer #2: No

---

## [Editor Report · Acceptance letter]

16 Apr 2021

PONE-D-21-06946R1 

Utility of hospitalization for elderly individuals affected by COVID-19 

Dear Dr. Solbiati:

I'm pleased to inform you that your manuscript has been deemed suitable for publication in PLOS ONE. Congratulations! Your manuscript is now with our production department. 

Kind regards, 

on behalf of

Dr. Francesco Di Gennaro 

Academic Editor

PLOS ONE